# Extended and Unscented Gaussian Processes

**Daniel M. Steinberg**
NICTA
daniel.steinberg@nicta.com.au

**Edwin V. Bonilla**
The University of New South Wales
e.bonilla@unsw.edu.au

## Abstract

We present two new methods for inference in Gaussian process (GP) models with general nonlinear likelihoods. Inference is based on a variational framework where a Gaussian posterior is assumed and the likelihood is linearized about the variational posterior mean using either a Taylor series expansion or statistical linearization. We show that the parameter updates obtained by these algorithms are equivalent to the state update equations in the iterative extended and unscented Kalman filters respectively, hence we refer to our algorithms as extended and unscented GPs. The unscented GP treats the likelihood as a 'black-box' by not requiring its derivative for inference, so it also applies to non-differentiable likelihood models. We evaluate the performance of our algorithms on a number of synthetic inversion problems and a binary classification dataset.

## 1 Introduction

Nonlinear inversion problems, where we wish to infer the latent inputs to a system given observations of its output and the system's forward-model, have a long history in the natural sciences, dynamical modeling and estimation. An example is the robot-arm inverse kinematics problem. We wish to infer how to drive the robot's joints (i.e. joint torques) in order to place the end-effector in a particular position, given we can measure its position and know the forward kinematics of the arm. Most of the existing algorithms either estimate the system inputs at a particular point in time like the Levenberg-Marquardt algorithm [1], or in a recursive manner such as the extended and unscented Kalman filters (EKF, UKF) [2].

In many inversion problems we have a continuous process; a smooth trajectory of a robot arm for example. Non-parametric regression techniques like Gaussian processes [3] seem applicable, and have been used in linear inversion problems [4]. Similarly, Gaussian processes have been used to learn inverse kinematics and predict the motion of a dynamical system such as robot arms [3, 5] and a human's gait [6, 7, 8]. However, in [3, 5] the inputs (torques) to the system are observable (not latent) and are used to train the GPs. Whereas [7, 8] are not concerned with inference over the original latent inputs, but rather they want to find a low dimensional representation of high dimensional outputs for prediction using Gaussian process latent variable models [6]. In this paper we introduce inference algorithms for GPs that can infer and predict the original latent inputs to a system, without having to be explicitly trained on them.

If we do not need to infer the latent inputs to a system it is desirable to still incorporate domain/system specific information into an algorithm in terms of a likelihood model specific to the task at hand. For example, non-parametric classification or robust regression problems. In these situations it is useful to have an inference procedure that does not require re-derivation for each new likelihood model without having to resort to MCMC. An example of this is the variational algorithm presented in [9] for factorizing likelihood models. In this model, the expectations arising from the use of arbitrary (non-conjugate) likelihoods are only one-dimensional, and so they can be easily evaluated using sampling techniques or quadrature. We present two alternatives to this algorithm that are also underpinned by variational principles but are based on linearizing the

nonlinear likelihood models about the posterior mean. These methods are straight-forwardly applicable to non-factorizing likelihoods and would retain computational efficiency, unlike [9] which would require evaluation of multidimensional intractable integrals. One of our algorithms, based on statistical linearization, does not even require derivatives of the likelihood model (like [9]) and so non-differentiable likelihoods can be incorporated.

Initially we formulate our models in §2 for the *finite* Gaussian case because the linearization methods are more general and comparable with existing algorithms. In fact we show we can derive the update steps of the iterative EKF [10] and similar updates to the iterative UKF [11] using our variational inference procedures. Then in §3 we specifically derive a factorizing likelihood Gaussian process model using our framework, which we use for experiments in §4.

## 2 Variational Inference in Nonlinear Gaussian Models with Linearization

Given some observable quantity $\mathbf{y} \in \mathbb{R}^d$, and a likelihood model for the system of interest, in many situations it is desirable to reason about the latent input to the system, $\mathbf{f} \in \mathbb{R}^D$, that generated the observations. Finding these inputs is an inversion problem and in a probabilistic setting it can be cast as an application of Bayes' rule. The following forms are assumed for the prior and likelihood:

$$p(\mathbf{f}) = \mathcal{N}(\mathbf{f}|\boldsymbol{\mu}, \mathbf{K}) \quad \text{and} \quad p(\mathbf{y}|\mathbf{f}) = \mathcal{N}(\mathbf{y}|g(\mathbf{f}), \boldsymbol{\Sigma}), \tag{1}$$

where $g(\cdot) : \mathbb{R}^D \to \mathbb{R}^d$ is a nonlinear function or forward model. Unfortunately the marginal likelihood, $p(\mathbf{y})$, is intractable as the nonlinear function makes the likelihood and prior non-conjugate. This also makes the posterior $p(\mathbf{f}|\mathbf{y})$, which is the solution to the inverse problem, intractable to evaluate. So, we choose to approximate the posterior with variational inference [12].

### 2.1 Variational Approximation

Using variational inference procedures we can put a lower bound on the log-marginal likelihood using Jensen's inequality,

$$\log p(\mathbf{y}) \geq \int q(\mathbf{f}) \log \frac{p(\mathbf{y}|\mathbf{f}) \, p(\mathbf{f})}{q(\mathbf{f})} d\mathbf{f}, \tag{2}$$

with equality iff $\mathrm{KL}[q(\mathbf{f}) \| p(\mathbf{f}|\mathbf{y})] = 0$, and where $q(\mathbf{f})$ is an approximation to the true posterior, $p(\mathbf{f}|\mathbf{y})$. This lower bound is often referred to as 'free energy', and can be re-written as follows

$$\mathcal{F} = \langle \log p(\mathbf{y}|\mathbf{f}) \rangle_{q\mathbf{f}} - \mathrm{KL}[q(\mathbf{f}) \| p(\mathbf{f})], \tag{3}$$

where $\langle \cdot \rangle_{q\mathbf{f}}$ is an expectation with respect to the variational posterior, $q(\mathbf{f})$. We assume the posterior takes a Gaussian form, $q(\mathbf{f}) = \mathcal{N}(\mathbf{f}|\mathbf{m}, \mathbf{C})$, so we can evaluate the expectation and KL term in (3),

$$\langle \log p(\mathbf{y}|\mathbf{f}) \rangle_{q\mathbf{f}} = -\frac{1}{2} \left[ D \log 2\pi + \log |\boldsymbol{\Sigma}| + \left\langle (\mathbf{y} - g(\mathbf{f}))^\top \boldsymbol{\Sigma}^{-1} (\mathbf{y} - g(\mathbf{f})) \right\rangle_{q\mathbf{f}} \right], \tag{4}$$

$$\mathrm{KL}[q(\mathbf{f}) \| p(\mathbf{f})] = \frac{1}{2} \left[ \mathrm{tr}(\mathbf{K}^{-1}\mathbf{C}) + (\boldsymbol{\mu} - \mathbf{m})^\top \mathbf{K}^{-1} (\boldsymbol{\mu} - \mathbf{m}) - \log |\mathbf{C}| + \log |\mathbf{K}| - D \right]. \tag{5}$$

where the expectation involving $g(\cdot)$ may be intractable. One method of dealing with these expectations is presented in [9] by assuming that the likelihood factorizes across observations. Here we provide two alternatives based on linearizing $g(\cdot)$ about the posterior mean, $\mathbf{m}$.

### 2.2 Parameter Updates

To find the optimal posterior mean, $\mathbf{m}$, we need to find the derivative,

$$\frac{\partial \mathcal{F}}{\partial \mathbf{m}} = -\frac{1}{2} \frac{\partial}{\partial \mathbf{m}} \left\langle (\boldsymbol{\mu} - \mathbf{f})^\top \mathbf{K}^{-1} (\boldsymbol{\mu} - \mathbf{f}) + (\mathbf{y} - g(\mathbf{f}))^\top \boldsymbol{\Sigma}^{-1} (\mathbf{y} - g(\mathbf{f})) \right\rangle_{q\mathbf{f}}, \tag{6}$$

where all terms in $\mathcal{F}$ independent of $\mathbf{m}$ have been dropped, and we have placed the quadratic and trace terms from the KL component in Equation (5) back into the expectation. We can represent this as an augmented Gaussian,

$$\frac{\partial \mathcal{F}}{\partial \mathbf{m}} = -\frac{1}{2} \frac{\partial}{\partial \mathbf{m}} \left\langle (\mathbf{z} - h(\mathbf{f}))^\top \mathbf{S}^{-1} (\mathbf{z} - h(\mathbf{f})) \right\rangle_{q\mathbf{f}}, \tag{7}$$

where

$$\mathbf{z} = \begin{bmatrix} \mathbf{y} \\ \boldsymbol{\mu} \end{bmatrix}, \quad h(\mathbf{f}) = \begin{bmatrix} g(\mathbf{f}) \\ \mathbf{f} \end{bmatrix}, \quad \mathbf{S} = \begin{bmatrix} \boldsymbol{\Sigma} & \mathbf{0} \\ \mathbf{0} & \mathbf{K} \end{bmatrix}. \tag{8}$$

Now we can see solving for $\mathbf{m}$ is essentially a nonlinear least squares problem, but about the expected posterior value of $\mathbf{f}$. Even without the expectation, there is no closed form solution to $\partial \mathcal{F} / \partial \mathbf{m} = 0$. However, we can use an iterative Newton method to find $\mathbf{m}$. It begins with an initial guess, $\mathbf{m}_0$, then proceeds with the iterations,

$$\mathbf{m}_{k+1} = \mathbf{m}_k - \alpha \left( \nabla_{\mathbf{m}} \nabla_{\mathbf{m}} \mathcal{F} \right)^{-1} \nabla_{\mathbf{m}} \mathcal{F}, \tag{9}$$

for some step length, $\alpha \in (0, 1]$. Though evaluating $\nabla_{\mathbf{m}} \mathcal{F}$ is still intractable because of the nonlinear term within the expectation in Equation (7). If we linearize $g(\mathbf{f})$, we can evaluate the expectation,

$$g(\mathbf{f}) \approx \mathbf{A} \mathbf{f} + \mathbf{b}, \tag{10}$$

for some linearization matrix $\mathbf{A} \in \mathbb{R}^{d \times D}$ and an intercept term $\mathbf{b} \in \mathbb{R}^d$. Using this we get,

$$\nabla_{\mathbf{m}} \mathcal{F} \approx \mathbf{A}^\top \boldsymbol{\Sigma}^{-1} (\mathbf{y} - \mathbf{A} \mathbf{m} - \mathbf{b}) + \mathbf{K}^{-1} (\boldsymbol{\mu} - \mathbf{m}) \quad \text{and} \quad \nabla_{\mathbf{m}} \nabla_{\mathbf{m}} \mathcal{F} \approx -\mathbf{K}^{-1} - \mathbf{A}^\top \boldsymbol{\Sigma}^{-1} \mathbf{A}. \tag{11}$$

Substituting (11) into (9) and using the Woodbury identity we can derive the iterations,

$$\mathbf{m}_{k+1} = (1 - \alpha) \mathbf{m}_k + \alpha \boldsymbol{\mu} + \alpha \mathbf{H}_k (\mathbf{y} - \mathbf{b}_k - \mathbf{A}_k \boldsymbol{\mu}), \tag{12}$$

where $\mathbf{H}_k$ is usually referred to as a "Kalman gain" term,

$$\mathbf{H}_k = \mathbf{K} \mathbf{A}_k^\top \left( \boldsymbol{\Sigma} + \mathbf{A}_k \mathbf{K} \mathbf{A}_k^\top \right)^{-1}, \tag{13}$$

and we have assumed that the linearization $\mathbf{A}_k$ and intercept, $\mathbf{b}_k$ are in some way dependent on the iteration. We can find the posterior covariance by setting $\partial \mathcal{F} / \partial \mathbf{C} = 0$ where,

$$\frac{\partial \mathcal{F}}{\partial \mathbf{C}} = -\frac{1}{2} \frac{\partial}{\partial \mathbf{C}} \left\langle (\mathbf{z} - h(\mathbf{f}))^\top \mathbf{S}^{-1} (\mathbf{z} - h(\mathbf{f})) \right\rangle_{q\mathbf{f}} + \frac{1}{2} \frac{\partial}{\partial \mathbf{C}} \log |\mathbf{C}|. \tag{14}$$

Again we do not have an analytic solution, so we once more apply the approximation (10) to get,

$$\mathbf{C} = \left[ \mathbf{K}^{-1} + \mathbf{A}^\top \boldsymbol{\Sigma}^{-1} \mathbf{A} \right]^{-1} = (\mathbf{I}_D - \mathbf{H} \mathbf{A}) \mathbf{K}, \tag{15}$$

where we have once more made use of the Woodbury identity and also the *converged* values of $\mathbf{A}$ and $\mathbf{H}$. At this point it is also worth noting the relationship between Equations (15) and (11).

## 2.3 Taylor Series Linearization

Now we need to find expressions for the linearization terms $\mathbf{A}$ and $\mathbf{b}$. One method is to use a first order Taylor Series expansion to linearize $g(\cdot)$ about the last calculation of the posterior mean, $\mathbf{m}_k$,

$$g(\mathbf{f}) \approx g(\mathbf{m}_k) + \mathbf{J}_{\mathbf{m}_k} (\mathbf{f} - \mathbf{m}_k), \tag{16}$$

where $\mathbf{J}_{\mathbf{m}_k}$ is the Jacobian $\partial g(\mathbf{m}_k) / \partial \mathbf{m}_k$. By linearizing the function in this way we end up with a *Gauss-Newton* optimization procedure for finding $\mathbf{m}$. Equating coefficients with (10),

$$\mathbf{A} = \mathbf{J}_{\mathbf{m}_k}, \qquad \mathbf{b} = g(\mathbf{m}_k) - \mathbf{J}_{\mathbf{m}_k} \mathbf{m}_k, \tag{17}$$

and then substituting these values into Equations (12) – (15) we get,

$$\mathbf{m}_{k+1} = (1 - \alpha) \mathbf{m}_k + \alpha \boldsymbol{\mu} + \alpha \mathbf{H}_k (\mathbf{y} - g(\mathbf{m}_k) + \mathbf{J}_{\mathbf{m}_k} (\mathbf{m}_k - \boldsymbol{\mu})), \tag{18}$$

$$\mathbf{H}_k = \mathbf{K} \mathbf{J}_{\mathbf{m}_k}^\top \left( \boldsymbol{\Sigma} + \mathbf{J}_{\mathbf{m}_k} \mathbf{K} \mathbf{J}_{\mathbf{m}_k}^\top \right)^{-1}, \tag{19}$$

$$\mathbf{C} = (\mathbf{I}_D - \mathbf{H} \mathbf{J}_{\mathbf{m}}) \mathbf{K}. \tag{20}$$

Here $\mathbf{J}_{\mathbf{m}}$ and $\mathbf{H}$ without the $k$ subscript are constructed about the converged posterior, $\mathbf{m}$.

**Remark 1** *A single step of the iterated extended Kalman filter [10, 11] corresponds to an update in our variational framework when using the Taylor series linearization of the non-linear forward model $g(\cdot)$ around the posterior mean.*

Having derived the updates in our variational framework, the proof of this is trivial by making $\alpha = 1$, and using Equations (18) – (20) as the iterative updates.

## 2.4 Statistical Linearization

Another method for linearizing $g(\cdot)$ is *statistical linearization* (see e.g. [13]), which finds a least squares best fit to $g(\cdot)$ about a point. The advantage of this method is that it does not require derivatives $\partial g(\mathbf{f})/\partial \mathbf{f}$. To obtain the fit, multiple observations of the forward model output for different input points are required. Hence, the key question is where to evaluate our forward model so as to obtain representative samples to carry out the linearization. One method of obtaining these points is the unscented transform [2], which defines $2D + 1$ 'sigma' points,

$$\mathcal{M}_0 = \mathbf{m}, \tag{21}$$

$$\mathcal{M}_i = \mathbf{m} + \left( \sqrt{(D + \kappa)\, \mathbf{C}} \right)_i \quad \text{for} \quad i = 1 \dots D, \tag{22}$$

$$\mathcal{M}_i = \mathbf{m} - \left( \sqrt{(D + \kappa)\, \mathbf{C}} \right)_i \quad \text{for} \quad i = D + 1 \dots 2D, \tag{23}$$

$$\mathcal{Y}_i = g(\mathcal{M}_i), \tag{24}$$

for a free parameter $\kappa$. Here $(\sqrt{\cdot})_i$ refers to columns of the matrix square root, we follow [2] and use the Cholesky decomposition. Unlike the usual unscented transform, which uses the prior to create the sigma points, here we have used the posterior because of the expectation in Equation (7). Using these points we can define the following statistics,

$$\bar{\mathbf{y}} = \sum_{i=0}^{2D} w_i \mathcal{Y}_i, \qquad \mathbf{\Gamma_{ym}} = \sum_{i=0}^{2D} w_i \left( \mathcal{Y}_i - \bar{\mathbf{y}} \right) \left( \mathcal{M}_i - \mathbf{m} \right)^{\top}, \tag{25}$$

$$w_0 = \frac{\kappa}{D + \kappa}, \qquad w_i = \frac{1}{2\,(D + \kappa)} \quad \text{for} \quad i = 1 \dots 2D. \tag{26}$$

According to [2] various settings of $\kappa$ can capture information about the higher order moments of the distribution of $\mathbf{y}$; or setting $\kappa = 0.5$ yields uniform weights. To find the linearization coefficients statistical linearization solves the following objective,

$$\underset{\mathbf{A},\mathbf{b}}{\text{argmin}} \sum_{i=0}^{2D} \| \mathcal{Y}_i - (\mathbf{A}\mathcal{M}_i + \mathbf{b}) \|_2^2. \tag{27}$$

This is simply linear least-squares and has the solution [13]:

$$\mathbf{A} = \mathbf{\Gamma_{ym}} \mathbf{C}^{\text{-1}}, \qquad \mathbf{b} = \bar{\mathbf{y}} - \mathbf{Am}. \tag{28}$$

Substituting $\mathbf{b}$ back into Equation (12), we obtain,

$$\mathbf{m}_{k+1} = (1 - \alpha)\, \mathbf{m}_k + \alpha \boldsymbol{\mu} + \alpha \mathbf{H}_k \left( \mathbf{y} - \bar{\mathbf{y}}_k + \mathbf{A}_k \left( \mathbf{m}_k - \boldsymbol{\mu} \right) \right). \tag{29}$$

Here $\mathbf{H}_k$, $\mathbf{A}_k$ and $\bar{\mathbf{y}}_k$ have been evaluated using the statistics from the $k$th iteration. This implies that the posterior covariance, $\mathbf{C}_k$, is now estimated at every iteration of (29) since we use it to form $\mathbf{A}_k$ and $\mathbf{b}_k$. $\mathbf{H}_k$ and $\mathbf{C}_k$ have the same form as Equations (13) and (15) respectively.

**Remark 2** *A single step of the iterated unscented sigma-point Kalman filter (iSPKF, [11]) can be seen as an ad hoc approximation to an update in our statistically linearized variational framework.*

Equations (29) and (15) are equivalent to the equations for a single update of the iterated sigma-point Kalman filter (iSPKF) for $\alpha = 1$, except for the term $\bar{\mathbf{y}}_k$ appearing in Equation (29) as opposed to $g(\mathbf{m}_k)$. The main difference is that we have derived our updates from variational principles. These updates are also more similar to the regular recursive unscented Kalman filter [2], and statistically linearized recursive least squares [13].

## 2.5 Optimizing the Posterior

Because of the expectations involving an arbitrary function in Equation (4), no analytical solution exists for the lower bound on the marginal likelihood, $\mathcal{F}$. We can use our approximation (10) again,

$$\mathcal{F} \approx -\frac{1}{2} \Bigg[ D \log 2\pi + \log |\mathbf{\Sigma}| - \log |\mathbf{C}| + \log |\mathbf{K}| + (\boldsymbol{\mu} - \mathbf{m})^{\top} \mathbf{K}^{\text{-1}} (\boldsymbol{\mu} - \mathbf{m})$$

$$+ (\mathbf{y} - \mathbf{Am} - \mathbf{b})^{\top} \mathbf{\Sigma}^{\text{-1}} (\mathbf{y} - \mathbf{Am} - \mathbf{b}) \Bigg]. \tag{30}$$

Here the trace term from Equation (5) has cancelled with a trace term from the expected likelihood, $\text{tr}\big(\mathbf{A}^\top \boldsymbol{\Sigma}^{\text{-1}} \mathbf{A}\mathbf{C}\big) = D - \text{tr}\big(\mathbf{K}^{\text{-1}}\mathbf{C}\big)$, once we have linearized $g(\cdot)$ and substituted (15). Unfortunately this approximation is no longer a lower bound on the log marginal likelihood in general. In practice we only calculate this approximation $\mathcal{F}$ if we need to optimize some model hyperparameters, like for a Gaussian process as described in §3. When optimizing $\mathbf{m}$, the only terms of $\mathcal{F}$ dependent on $\mathbf{m}$ in the Taylor series linearization case are,

$$-\frac{1}{2} \left(\mathbf{y} - g(\mathbf{m})\right)^\top \boldsymbol{\Sigma}^{\text{-1}} \left(\mathbf{y} - g(\mathbf{m})\right) - \frac{1}{2} \left(\boldsymbol{\mu} - \mathbf{m}\right)^\top \mathbf{K}^{\text{-1}} \left(\boldsymbol{\mu} - \mathbf{m}\right). \qquad (31)$$

This is also the *maximum a-posteriori* objective. A global convergence proof exists for this objective when optimized by a Gauss-Newton procedure, like our Taylor series linearization algorithm, under some conditions on the Jacobians, see [14, p255]. No such guarantees exist for statistical linearization, though monitoring (31) works well in practice (see the experiment in §4.1).

A line search could be used to select an optimal value for the step length, $\alpha$ in Equation (12). However, we find that setting $\alpha = 1$, and then successively multiplying $\alpha$ by some number in $(0, 1)$ until the MAP objective (31) decreases, or some maximum number of iterations is exceeded is fast and works well in practice. If the maximum number of iterations is exceeded we call this a 'diverge' condition, and terminate the search for $\mathbf{m}$ (and return the last good value). This only tends to happen for statistical linearization, but does not tend to impact the algorithms performance since we always make sure to improve (approximate) $\mathcal{F}$.

## 3  Variational Inference in Gaussian Process Models with Linearization

We now present two inference methods for Gaussian Process (GP) models [3] with arbitrary nonlinear likelihoods using the framework presented previously. Both Gaussian process models have the following likelihood and prior,

$$\mathbf{y} \sim \mathcal{N}\big(g(\mathbf{f}), \sigma^2 \mathbf{I}_N\big), \qquad \mathbf{f} \sim \mathcal{N}(\mathbf{0}, \mathbf{K}). \qquad (32)$$

Here $\mathbf{y} \in \mathbb{R}^N$ are the $N$ noisy observed values of the transformed latent function, $g(\mathbf{f})$, and $\mathbf{f} \in \mathbb{R}^N$ is the latent function we are interested in inferring. $\mathbf{K} \in \mathbb{R}^{N \times N}$ is the kernel matrix, where each element $k_{ij} = k(\mathbf{x}_i, \mathbf{x}_j)$ is the result of applying a kernel function to each input, $\mathbf{x} \in \mathbb{R}^P$, in a pairwise manner. It is also important to note that the likelihood noise model is *isotropic* with a variance of $\sigma^2$. This is not a necessary condition, and we can use a correlated noise likelihood model, however the factorized likelihood case is still useful and provides some computational benefits.

As before, we make the approximation that the posterior is Gaussian, $q(\mathbf{f}|\mathbf{m}, \mathbf{C}) = \mathcal{N}(\mathbf{f}|\mathbf{m}, \mathbf{C})$ where $\mathbf{m} \in \mathbb{R}^N$ is the mean posterior latent function, and $\mathbf{C} \in \mathbb{R}^{N \times N}$ is the posterior covariance. Since the likelihood is isotropic and factorizes over the $N$ observations we have the following expectation under our variational inference framework:

$$\langle \log p(\mathbf{y}|\mathbf{f}) \rangle_{q\mathbf{f}} = -\frac{N}{2} \log 2\pi\sigma^2 - \frac{1}{2\sigma^2} \sum_{n=1}^{N} \left\langle (y_n - g(f_n))^2 \right\rangle_{q f_n}.$$

As a consequence, the linearization is one-dimensional, that is $g(f_n) \approx a_n f_n + b_n$. Using this we can derive the approximate gradients,

$$\nabla_{\mathbf{m}} \mathcal{F} \approx \frac{1}{\sigma^2} \mathbf{A} \left(\mathbf{y} - \mathbf{A}\mathbf{m} - \mathbf{b}\right) - \mathbf{K}^{\text{-1}}\mathbf{m}, \qquad \nabla_{\mathbf{m}} \nabla_{\mathbf{m}} \mathcal{F} \approx -\mathbf{K}^{\text{-1}} - \mathbf{A}\boldsymbol{\Lambda}^{\text{-1}}\mathbf{A}, \qquad (33)$$

where $\mathbf{A} = \text{diag}([a_1, \ldots, a_N])$ and $\boldsymbol{\Lambda} = \text{diag}\big([\sigma^2, \ldots, \sigma^2]\big)$. Because of the factorizing likelihood we obtain $\mathbf{C}^{\text{-1}} = \mathbf{K}^{\text{-1}} + \mathbf{A}\boldsymbol{\Lambda}^{\text{-1}}\mathbf{A}$, that is, the inverse posterior covariance is just the prior inverse covariance, but with a modified diagonal. This means if we were to use this inverse parameterization of the Gaussian, which is also used in [9], we would only have to infer $2N$ parameters (instead of $N + N(N+1)/2$). We can obtain the iterative steps for $\mathbf{m}$ straightforwardly:

$$\mathbf{m}_{k+1} = (1 - \alpha)\,\mathbf{m}_k + \alpha \mathbf{H}_k \left(\mathbf{y} - \mathbf{b}_k\right), \quad \text{where} \quad \mathbf{H}_k = \mathbf{K}\mathbf{A}_k \left(\boldsymbol{\Lambda} + \mathbf{A}_k \mathbf{K} \mathbf{A}_k\right)^{\text{-1}}, \qquad (34)$$

and also an expression for posterior covariance,

$$\mathbf{C} = (\mathbf{I}_N - \mathbf{H}\mathbf{A})\mathbf{K}. \qquad (35)$$

The values for $a_n$ and $b_n$ for the linearization methods are,

$$\text{Taylor}: \quad a_n = \frac{\partial g(m_n)}{\partial m_n}, \quad b_n = g(m_n) - \frac{\partial g(m_n)}{\partial m_n} m_n, \tag{36}$$

$$\text{Statistical}: \quad a_n = \frac{\Gamma_{my,n}}{C_{nn}}, \quad b_n = \bar{y}_n - a_n m_n. \tag{37}$$

$C_{nn}$ is the $n$th diagonal element of $\mathbf{C}$, and $\Gamma_{my,n}$ and $\bar{y}_n$ are scalar versions of Equations (21) – (26). The sigma points for each observation, $n$, are $\mathcal{M}_n = \{m_n, \ m_n + \sqrt{(1+\kappa)\,C_{nn}}, \ m_n - \sqrt{(1+\kappa)\,C_{nn}}\}$. We refer to the Taylor series linearized GP as the *extended* GP (EGP), and the statistically linearized GP as the *unscented* GP (UGP).

## 3.1 Prediction

The predictive distribution of a latent value, $f^*$, given a query point, $\mathbf{x}^*$, requires the marginalization $\int p(f^*|\mathbf{f})\, q(\mathbf{f}|\mathbf{m}, \mathbf{C})\, d\mathbf{f}$, where $p(f^*|\mathbf{f})$ is a regular predictive GP. This gives $f^* \sim \mathcal{N}(m^*, C^*)$, and,

$$m^* = \mathbf{k}^{*\top}\mathbf{K}^{-1}\mathbf{m}, \qquad C^* = k^{**} - \mathbf{k}^{*\top}\mathbf{K}^{-1}\left[\mathbf{I}_N - \mathbf{C}\mathbf{K}^{-1}\right]\mathbf{k}^*, \tag{38}$$

where $k^{**} = k(\mathbf{x}^*, \mathbf{x}^*)$ and $\mathbf{k}^* = [k(\mathbf{x}_1, \mathbf{x}^*), \ldots, k(\mathbf{x}_N, \mathbf{x}^*)]^\top$. We can also find the predicted observations, $\bar{y}^*$ by evaluating the one-dimensional integral,

$$\bar{y}^* = \langle y^* \rangle_{qf^*} = \int g(f^*)\, \mathcal{N}(f^*|m^*, C^*)\, df^*, \tag{39}$$

for which we use quadrature. Alternatively, if we were to use the UGP we can use another application of the unscented transform to approximate the predictive distribution $y^* \sim \mathcal{N}(\bar{y}^*, \sigma_{y^*}^2)$ where,

$$\bar{y}^* = \sum_{i=0}^{2} w_i \mathcal{M}_i^*, \qquad \sigma_{y^*}^2 = \sum_{i=0}^{2} w_i \left(\mathcal{Y}_i^* - \bar{y}^*\right)^2. \tag{40}$$

This works well in practice, see Figure 1 for a demonstration.

## 3.2 Learning the Linearized GPs

Learning the extended and unscented GPs consists of an inner and outer loop. Much like the Laplace approximation for binary Gaussian Process classifiers [3], the inner loop is for learning the posterior mean, $\mathbf{m}$, and the outer loop is to optimize the likelihood parameters (e.g. the variance $\sigma^2$) and kernel hyperparameters, $k(\cdot, \cdot|\boldsymbol{\theta})$. The dominant computational cost in learning the parameters is the inversion in Equation (34), and so the computational complexity of the EGP and UGP is about the same as for the Laplace GP approximation. To learn the kernel hyperparameters and $\sigma^2$ we use numerical techniques to find the gradients, $\partial\mathcal{F}/\partial\boldsymbol{\theta}$, for both the algorithms, where $\mathcal{F}$ is approximated,

$$\mathcal{F} \approx -\frac{1}{2}\left[N\log 2\pi\sigma^2 - \log|\mathbf{C}| + \log|\mathbf{K}| + \mathbf{m}^\top\mathbf{K}^{-1}\mathbf{m} + \frac{1}{\sigma^2}(\mathbf{y} - \mathbf{A}\mathbf{m} - \mathbf{b})^\top(\mathbf{y} - \mathbf{A}\mathbf{m} - \mathbf{b})\right]. \tag{41}$$

Specifically we use derivative-free optimization methods (e.g. BOBYQA) from the NLopt library [15], which we find fast and effective. This also has the advantage of not requiring knowledge of $\partial g(f)/\partial f$ or higher order derivatives for any implicit gradient dependencies between $\mathbf{f}$ and $\boldsymbol{\theta}$.

# 4 Experiments

## 4.1 Toy Inversion Problems

In this experiment we generate 'latent' function data from $\mathbf{f} \sim \mathcal{N}(\mathbf{0}, \mathbf{K})$ where a Matérn $\frac{5}{2}$ kernel function is used with amplitude $\sigma_{m52} = 0.8$, length scale $l_{m52} = 0.6$ and $\mathbf{x} \in \mathbb{R}$ are uniformly spaced between $[-2\pi, 2\pi]$ to build $\mathbf{K}$. Observations used to test and train the GPs are then generated as $\mathbf{y} = g(\mathbf{f}) + \epsilon$ where $\epsilon \sim \mathcal{N}(0, 0.2^2)$. 1000 points are generated in this way, and we use 5-fold cross validation to train (200 points) and test (800 points) the GPs. We use standardized mean

Table 1: The negative log predictive density (NLPD) and the standardized mean squared error (SMSE) on test data for various differentiable forward models. Lower values are better for both measures. The predicted $f^*$ and $y^*$ are the same for $g(\mathbf{f}) = \mathbf{f}$, so we do not report $y^*$ in this case.

| $g(\mathbf{f})$ | Algorithm | NLPD $f^*$ | | SMSE $f^*$ | | SMSE $y^*$ | |
|---|---|---|---|---|---|---|---|
| | | mean | std. | mean | std. | mean | std. |
| $\mathbf{f}$ | UGP | -0.90046 | 0.06743 | 0.01219 | 0.00171 | – | – |
| | EGP | -0.89908 | 0.06608 | 0.01224 | 0.00178 | – | – |
| | [9] | -0.27590 | 0.06884 | 0.01249 | 0.00159 | – | – |
| | GP | **-0.90278** | 0.06988 | **0.01211** | 0.00160 | – | – |
| $\mathbf{f}^3 + \mathbf{f}^2 + \mathbf{f}$ | UGP | **-0.23622** | 1.72609 | 0.01534 | 0.00202 | **0.02184** | 0.00525 |
| | EGP | -0.22325 | 1.76231 | **0.01518** | 0.00203 | **0.02184** | 0.00528 |
| | [9] | -0.14559 | 0.04026 | 0.06733 | 0.01421 | 0.02686 | 0.00266 |
| $\exp(\mathbf{f})$ | UGP | -0.75475 | 0.32376 | **0.13860** | 0.04833 | **0.03865** | 0.00403 |
| | EGP | **-0.75706** | 0.32051 | 0.13971 | 0.04842 | 0.03872 | 0.00411 |
| | [9] | -0.08176 | 0.10986 | 0.17614 | 0.04845 | 0.05956 | 0.01070 |
| $\sin(\mathbf{f})$ | UGP | **-0.59710** | 0.22861 | **0.03305** | 0.00840 | 0.11513 | 0.00521 |
| | EGP | -0.59705 | 0.21611 | 0.03480 | 0.00791 | **0.11478** | 0.00532 |
| | [9] | -0.04363 | 0.03883 | 0.05913 | 0.01079 | 0.11890 | 0.00652 |
| $\tanh(2\mathbf{f})$ | UGP | **0.01101** | 0.60256 | **0.15703** | 0.06077 | **0.08767** | 0.00292 |
| | EGP | 0.57403 | 1.25248 | 0.18739 | 0.07869 | 0.08874 | 0.00394 |
| | [9] | 0.15743 | 0.14663 | 0.16049 | 0.04563 | 0.09434 | 0.00425 |

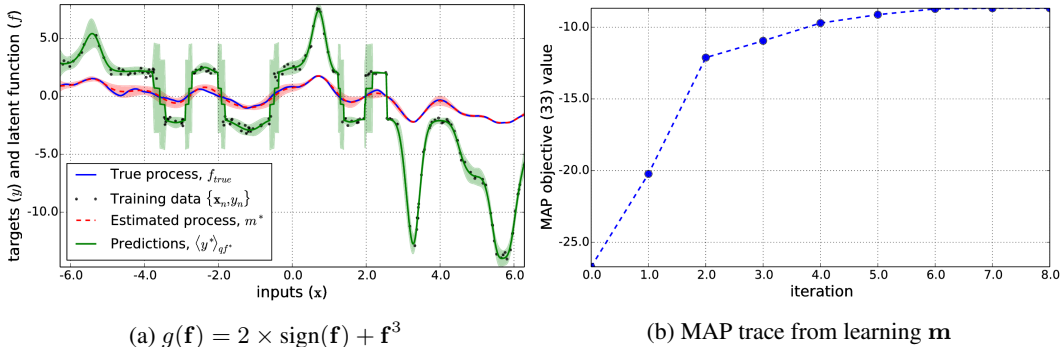

(a) $g(\mathbf{f}) = 2 \times \text{sign}(\mathbf{f}) + \mathbf{f}^3$

(b) MAP trace from learning $\mathbf{m}$

Figure 1: Learning the UGP with a non-differentiable forward model in (a), and a corresponding trace from the MAP objective function used to learn $\mathbf{m}$ is shown in (b). The optimization shown terminated because of a 'divergence' condition, though the objective function value has still improved.

squared error (SMSE) to test the predictions with the held out data in both the latent and observed spaces. We also use average negative log predictive density (NLPD) on the latent test data, which is calculated as $-\frac{1}{N^*}\sum_n \log \mathcal{N}(f_n^*|m_n^*, C_n^*)$. All GP methods use Matérn $\frac{5}{2}$ covariance functions with the hyperparameters and $\sigma^2$ initialized at 1.0 and lower-bounded at 0.1 (and 0.01 for $\sigma^2$).

Table 1 shows results for multiple differentiable forward models, $g(\cdot)$. We test the EGP and UGP against the model in [9] – which uses 10,000 samples to evaluate the one dimensional expectations. Although this number of samples may seem excessive for these simple problems, our goal here is to have a competitive baseline algorithm. We also test against normal GP regression for a linear forward model, $g(\mathbf{f}) = \mathbf{f}$. In Figure 1 we show the results of the UGP using a forward model for which no derivative exists at the zero crossing points, as well as an objective function trace for learning the posterior mean. We use quadrature for the predictions in observation space in Table 1 and the unscented transform, Equation (40), for the predictions in Figure 1. Interestingly, there is almost no difference in performance between the EGP and UGP, even though the EGP has access to the derivatives of the forward models and the UGP does not. Both the UGP and EGP consistently outperformed [9] in terms of NLPD and SMSE, apart from the tanh experiment for inversion. In this experiment, the UGP had the best performance but the EGP was outperformed by [9].

Table 2: Classification performance on the USPS handwritten-digits dataset for numbers '3' and '5'. Lower values of the negative log probability (NLP) and error rate indicate better performance. The learned signal variance $\left(\sigma_{\text{se}}^2\right)$ and length scale $(l_{\text{se}})$ are also shown for consistency with [3, §3.7.3].

| Algorithm | NLP $y^*$ | Error rate (%) | $\log(\sigma_{\text{se}})$ | $\log(l_{\text{se}})$ |
|---|---|---|---|---|
| GP – Laplace | 0.11528 | 2.9754 | 2.5855 | 2.5823 |
| GP – EP | 0.07522 | 2.4580 | 5.2209 | 2.5315 |
| GP – VB | 0.10891 | 3.3635 | 0.9045 | 2.0664 |
| SVM (RBF) | 0.08055 | 2.3286 | – | – |
| Logistic Reg. | 0.11995 | 3.6223 | – | – |
| UGP | **0.07290** | **1.9405** | 1.5743 | 1.5262 |
| EGP | 0.08051 | 2.1992 | 2.9134 | 1.7872 |

## 4.2 Binary Handwritten Digit Classification

For this experiment we evaluate the EGP and UGP on a classification task. We are just interested in a probabilistic prediction of class labels, and not the values of the latent function. We use the USPS handwritten digits dataset with the task of distinguishing between '3' and '5' – this is the same experiment from [3, §3.7.3]. A logistic sigmoid is used as the forward model, $g(\cdot)$, in our algorithms. We test against Laplace, expectation propagation and variational Bayes logistic GP classifiers (from the GPML Matlab toolbox [3]), a support vector machine (SVM) with a radial basis kernel function (and probabilistic outputs [16]), and logistic regression (both from the scikit-learn python library [17]). A squared exponential kernel with amplitude $\sigma_{\text{se}}$ and length scale $l_{\text{se}}$ is used for the GPs in this experiment. We initialize these hyperparameters at 1.0, and put a lower bound of 0.1 on them. We initialize $\sigma^2$ and place a lower bound at $10^{-14}$ for the EGP and UGP (the optimized values are near or at this value). The hyperparameters for the SVM are learned using grid search with three-fold cross validation.

The results are summarized in Table 2, where we report the average Bernoulli negative log-probability (NLP), the error rate and the learned hyperparameter values for the GPs. Surprisingly, the UGP outperforms the other classifiers on this dataset, despite the other classifiers being specifically formulated for this task.

## 5 Conclusion and Discussion

We have presented a variational inference framework with linearization for Gaussian models with nonlinear likelihood functions, which we show can be used to derive updates for the extended and unscented Kalman filter algorithms, the iEKF and the iSPKF. We then generalize these results and develop two inference algorithms for Gaussian processes, the EGP and UGP. The UGP does not use derivatives of the nonlinear forward model, yet performs as well as the EGP for inversion and classification problems.

Our method is similar to the Warped GP (WGP) [18], however, we wish to infer the *full* posterior over the latent function $f$. The goal of the WGP is to infer a transformation of a non-Gaussian process observation to a space where a GP can be constructed. That is, the WGP is concerned with inferring an inverse function $g^{-1}(\cdot)$ so the transformed (latent) function is well modeled by a GP.

As future work we would like to create multi-task EGPs and UGPs. This would extend their applicability to inversion problems where the forward models have multiple inputs and outputs, such as inverse kinematics for dynamical systems.

#### Acknowledgments

This research was supported by the Science Industry Endowment Fund (RP 04-174) Big Data Knowledge Discovery project. We thank F. Ramos, L. McCalman, S. O'Callaghan, A. Reid and T. Nguyen for their helpful feedback. NICTA is funded by the Australian Government through the Department of Communications and the Australian Research Council through the ICT Centre of Excellence Program.

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
