[Reviews · NeurIPS 2014]

Submitted by Assigned_Reviewer_8

This is an interesting and well written paper using variational
approximations to deal with non-Gaussian likelihoods.

The ideas in the paper are very well explained and the connections to
the existing literature are good. The connections to the EKF and UKF
are interesting.

In section 1 and in section 3, it is claimed that the proposed
methodology would be applicable to problems with non-factorizing
likelihoods, but no further details are provided about this, and no
experiments deal with this case. Perhaps it would be clearer if these
notes were left to the conclusions (and outlook) final section?

The experimental section is good, including a fairly wide variety of
non-linear "g" functions. The comparison to ref [9] is good, however,
it would be nice to know *why* that method is being outperformed
(except in the tanh case). Can you provide any insight as to why and
when which method is the best approximation? The experimental section
could be further strengthened by including other competitive
algorithms in table 1: how would Laplace's method compare, and could
Expectation Propagation (EP) be applied?

The binary classification example is also good with good
comparisons. Since the likelihood used by the EGP and UGP methods in
non-standard (sigmoid + Gaussian noise) it would be relevant explain
what happens, ie, what is the inferred noise level, what happens with
the probability mass which exceeds the binary labels, can this just be
ignored? It is a bit unclear how the predictive class label is
defined... equation (42) just gives the predictive *mean*, what about
the predictive uncertainty, pls elaborate.

The method uses gradient free hyperparameter learning, which is not
the norm for comparable algorithms. Pls give a few more details about
this. In the examples provided only very few hyperparameters are
used... perhaps in this case gradient free hyperparameter optimization
is ok... would you expect it to work well if there were more (eg 10s)
of hyperparameters? Pls elaborate.

minor point: it may be helpful to state right away what is meant by
the term "non-linear likelihood" (when this became clear in section
2, I had to go back an re-read section 1).
Summary: This is a good, interesting and clearly written paper on the use of
the variational approximation framework to GP inference with
non-Gaussian likelihoods. The exposition of the method is good, and
the experimental section is clear and well presented. Good comparisons
are chosen; these could be strengthened by including other
competitors, eg Laplace's method and possibly EP. A good contribution
to the GP literature.

Submitted by Assigned_Reviewer_15

The authors present two similar methods for approximate inference in latent Gaussian models. They consider models of a particular form, where the likelihood is Gaussian after a nonlinear transform of the latent values. This model class leads them to consider linearisation of the transformation, which in turn leads to algorithms for variational updates which resembe the extended and unscented Kalman filter.

Pros
--
What an interesting read. I thouroughly enjoyed reviewing this paper. It presents the ideas clearly and thouroughly, I feel like I could implement these ideas quickly given the detail presented.

I think the authors should perhaps have made more of the particular model class they have chosen, which is really interesting. Table 1 shows that the method work well for a range of difference nonlinearities. It's a shame there's no mention of learning the nonlinear function anywhere (aside from the noise variance), I would think that would be of great interest.

Given the model class, I'm sure there is a relationship with the Warped GP work of Ed Snelson? Perhaps this is worth a mention?

Critisism
--
line 166. the authors describe their method as 'variational updates'. I think this is a litle misleading, since variational updates are guaranteed to increase the objective function, and thus reduce the KL. Here, this isn;t the case. Given the relationship with the unscented/extended kalman filter, It would be great to know a litle about the convergece properties of the approximations. Is convergence guaranteed?

Table 2 compares the negative log probability of the method with GP classification from the GPML toolbox. I think this is very misleading: the two methods are using different models, and so these numbers really cannot be compared. I strongly recommend removing this.

It's a shame that the method doesn't give gradient of the hyper-parameters for optimization. I suppose the authors are up-front about this, but I think this will put people off adopting the approach in general.

What's going on in Figure 1 (b)? this is for one particular inner loop? Seems like a relatively uninformative plot to me.

Table 1, first row, contains a wierd result, in that the GP method and the [9] method differ. I think in this case, posterior is Gaussian and the approximation in [9] is exact, yet the results are different to the GP result. What gives?

Typos
--
equations 8,9, missing a \partial symbol on the denominator
Summary: Summary
--
Excellent authorship and an entertaining paper. Would be improved by discussion of the convergence properties of the method and a compelling application of the model.

I feel that the comparison of the log likelihoods across two different models is unfair, and should be removed. the discrepancy between the exact GP and the variational method is very strange.

Prepared to increase my score if issues are addressed.

EDIT
===
The authors clarified the way they were performing the held-out density, I'm much happier with this now. Score increased accordingly.

Submitted by Assigned_Reviewer_43

The authors present a simple idea for approximating the variational lower bound on the marginal likelihood for GPs in the case of an intractable likelihood p(y | f) with a particular form --- the observations y equal an arbitrary nonlinear function g(f) with additive Gaussian noise. The lower bound requires solving an expectation involving the nonlinear function g(f), which can't be done analytically. The authors suggest linearising the function about the current estimate of the posterior mean on f. The paper contains two methods for performing the linearisation: by differentiation of g (as in the extended Kalman filter), and by use of `sigma points' (as in the unscented transform). The authors show how variational inference can then be carried out with their approximations along with learning of the GP hyperparameters, and making predictions at new test points. Empirical evaluation is provided by using their method for regression with 4 different nonlinear functions g(f), and for classification on the USPS dataset using a logistic likelihood.

Although the idea is fairly simple it is still a worthy addition to the family of approximate GP inference methods for intractable likelihoods. By approximating the lower bound the method does lose many of the nice statistical properties of variational inference and the paper could benefit from some discussion of this.

The paper is mostly well written and provides a clear exposition. I would like the authors to provide some disambiguation around the terms `input' and 'inverse model' however: in section 2 the latent variable f is referred to as the input, whereas in section 3 we now have x which is the input. This is compounded by the discussion of inverse models, by which the authors mean the mapping from the observations y to the latent variable f but which could easily be understood in the GP setting to mean the mapping from the `output' f (or y) to the `input' x. Other than this I find the paper to be eminently understandable.

The experiments seem a little suspect to me. There is very little discussion of the results presented in table 1, which the paper would benefit from as very few of the results appear to be significant, given the quoted standard deviations --- for NLL `exp(f)' and `sin(f)' might be significant but the polynomial and tanh functions don't appear to be. Is this also the case for the results in table 2? The error bars are not reported here. The authors should also compare, both theoretically and experimentally, to the variational classification method implemented in the GPML toolbox for the logistic likelihood.
Summary: The paper approximately computes the variational lower bound on GP inference with likelihoods of the form p(y |f) = N(g(f), S) for arbitrary, nonlinear functions g, by linearising g(f). The method is simple but still interesting and the paper is mostly clear and understandable although it would benefit from some deeper theoretical comparisons and discussion of the experiments.
Author Feedback
Author rebuttal: Thank you to the reviewers for thoroughly reading our paper and for providing constructive feedback. All of the suggestions for improving clarity of the text, discussions of results, and corrections will be incorporated into the final version.

R15-1: There is no mention of learning the nonlinear function, which would be of interest.

Parameters can be learned for this function (currently implemented), either using provided or numerical gradients. However, this is not the goal of the paper as we wish to find the posterior over the latent inputs to this function.

R15-2: I’m sure there is a relationship with Warped GPs?

While there are some similarities mechanistically with the warped GP, our objective is different in that we wish to infer the full posterior over the latent function, f. The goal of a warped GP is to infer a transformation of a non-Gaussian process observation to a space where a GP can be constructed. That is, given an “inverse” function f = g^-1(y, theta), learn theta and the underlying GP. We can add this to the introduction.

R15-3: Expresses concern that linearisation removes the guarantee of the variational objective (F) lower bounding LML. Also wants to know if convergence is guaranteed.

When eq 14 has not converged we cannot guarantee F is still a lower bound. However, in the case of the EGP, a Gauss-Newton procedure is used to find the optimal m (mentioned in section 2.3 and 2.5), which has a global convergence proof under some conditions on the Jacobians, see [14] p255. Hence, when we are using converged m in the EGP (and the Jacobian conditions are satisfied), evaluating F is a valid lower bound. We cannot make the same guarantees for the UGP (though it works well in practice), but searching for such guarantees may be interesting future work. The secant method may provide some insight into this, perhaps also the observation that as the spread of sigma points (kappa) approaches zero, the A matrix will approach the Jacobian of g. These points can be clarified in the text.

R15-4: It is not valid to compare the likelihoods from the GP methods with those from the GPML toolbox.

This is not the case, and we can clarify this point in the paper. Even though our method has a different likelihood to the methods in the GMPL toolbox, the expectation, E[y*], from our model is actually the expected probability of a label, i.e., our equation (42) is the same as (3.25) from [3] for the classification experiment. We then evaluate the log likelihood of the true labels given these expected label probabilities using a Bernoulli likelihood. This is also known as the negative log probability loss, and is a standard classification benchmark, see;
“Evaluating predictive uncertainty challenge”, Quinonero-Candela et al 2005.

R15-5: We do not give hyperparameter gradients for our method.

Unfortunately there is a strong coupling between the hyperparameters and posterior parameter gradients (in the EGP), and so we have found that numerical methods work well and are easily implementable in this situation. These methods may not work for e.g., high dimensional ARD kernel functions (or more than ~10 hyperparameters). Using automatic differentiation may be an interesting alternative for both the EGP and UGP.

R15-6: Why does the method [9] do badly, even in the normal GP setting?

We found that hyperparameter learning for [9] (essentially stochastic gradients) was not as effective as the numerical methods used by the other methods. In the identity forward model case if we were to set the hyperparameters to be the same for all algorithms we would expect the same results -- however this is not a particularly realistic setting so we did not include it in the experiment. We can clarify this in the text.

R43-1: By approximating the lower bound many of the statistical properties of variational inference are lost.

Please see R15-3:

R43-2: Very few of the results seem to be significant in Table 1, and there could be more discussion.

We feel that this statement is perhaps a little strong -- we would suggest that only the tanh(2f) forward models do not have a significantly better NLL than [9]. Furthermore, the UGP can be used in situations where [9] and the EGP cannot (fig 1a), without sacrificing performance in situations where all methods can be used. This is a significant result and is one of the main contributions of the paper.

R43-3: Error bars are not reported on Table 2.

We do not report error bars on this dataset because no cross validation is performed -- there is a standard training and test set that [6] has established. We can clarify this point in the paper.

R43-4: Compare to the variational method in the GPML toolbox.

Here are the results:
VB: av nll = 0.10891, err = 3.3635%, l_sig = 0.9045, l_l = 2.0664
This is comparable to the Laplace approximation, and worse than the UGP.
We remind the reviewers that the variational GP classifier, like EP, has to be specifically derived for each forward model (bounding the sigmoid in this case), so is not as general as our methods.

R8-1: Why is [9] being outperformed?

see R15-6.

R8-2: Explain more about the sigmoid + Gaussian likelihood, i.e. inferred noise level and probability mass outside the bounds [0, 1].

We have found that for both the EGP and UGP the likelihood noise is pushed towards 0 (within numerical limits and subject to local extrema in F -- mentioned in the paper), and so effectively the likelihood becomes a sigmoid within a dirac function, making the probability mass negligible outside [0, 1]. The predictive uncertainty can be obtained by using eq (43) (this makes less sense for classification) and has been used in fig 1a. We could also potentially evaluate the integral int (E[y*] - g(f*))^2 N(f*|m*, C*)df*. These points can be added to the paper.

R8-3: Please give more details of parameter learning.

see R15-5.